# Cellular and Humoral Immunity against Different SARS-CoV-2 Variants Is Detectable but Reduced in Vaccinated Kidney Transplant Patients

**DOI:** 10.3390/vaccines10081348

**Published:** 2022-08-18

**Authors:** Laura Thümmler, Anja Gäckler, Maren Bormann, Sandra Ciesek, Marek Widera, Hana Rohn, Neslinur Fisenkci, Mona Otte, Mira Alt, Ulf Dittmer, Peter A. Horn, Oliver Witzke, Adalbert Krawczyk, Monika Lindemann

**Affiliations:** 1Department of Infectious Diseases, West German Centre of Infectious Diseases, University Hospital Essen, University of Duisburg-Essen, 45147 Essen, Germany; 2Institute for Transfusion Medicine, University Hospital Essen, University of Duisburg-Essen, 45147 Essen, Germany; 3Department of Nephrology, University Hospital Essen, University of Duisburg-Essen, 45147 Essen, Germany; 4Institute of Medical Virology, University Hospital Frankfurt, 60590 Frankfurt am Main, Germany; 5Institute of Pharmaceutical Biology, Goethe-University, 60323 Frankfurt am Main, Germany; 6Fraunhofer Institute for Molecular Biology and Applied Ecology (IME), Branch Translational Medicine and Pharmacology, 60311 Frankfurt am Main, Germany; 7Institute for Virology, University Hospital Essen, University Duisburg-Essen, 45147 Essen, Germany

**Keywords:** ELISpot, T cells, vaccination

## Abstract

In kidney transplant (KTX) patients, immune responses after booster vaccination against SARS-CoV-2 are inadequately examined. We analyzed these patients a median of four months after a third/fourth vaccination and compared them to healthy controls. Cellular responses were analyzed by interferon-gamma (IFN-γ) and interleukin-2 (IL-2) ELISpot assays. Neutralizing antibody titers were assessed against SARS-CoV-2 D614G (wild type) and the variants alpha, delta, and omicron by a cell culture-based neutralization assay. Humoral immunity was also determined by a competitive fluorescence assay, using 11 different variants of SARS-CoV-2. Antibody ratios were measured by ELISA. KTX patients showed significantly lower SARS-CoV-2-specific IFN-γ responses after booster vaccination than healthy controls. However, SARS-CoV-2-specific IL-2 responses were comparable to the T cell responses of healthy controls. Cell culture-based neutralizing antibody titers were 1.3-fold higher in healthy controls for D614G, alpha, and delta, and 7.8-fold higher for omicron (*p* < 0.01). Healthy controls had approximately 2-fold higher concentrations of potential neutralizing antibodies against all 11 variants than KTX patients. However, more than 60% of the KTX patients displayed antibodies to variants of SARS-CoV-2. Thus, KTX patients should be partly protected, due to neutralizing antibodies to variants of SARS-CoV-2 or by cross-reactive T cells, especially those producing IL-2.

## 1. Introduction

Since the first appearance of severe acute respiratory syndrome coronavirus type 2 (SARS-CoV-2) in December 2019, more than 500 million people have been infected with SARS-CoV-2 and more than 6 million people have died from coronavirus disease 19 (COVID-19) (June 2022) [1].

Immunocompromised individuals, such as cancer patients, solid organ recipients, and individuals with comorbidities, have a higher mortality and morbidity rate from COVID-19 [2,3,4]. Individuals who belong to vulnerable groups benefit from vaccination against SARS-CoV-2 to protect themselves from infection. They can also be protected indirectly by vaccinating individuals around them, as this significantly reduces the risk of infection [5,6,7].

However, studies displayed only weak or no vaccination responses after SARS-CoV-2 infection and two mRNA vaccinations in immunosuppressed patients who also suffer more frequently from vaccine breakthrough infection [8,9,10]. Previous studies have shown that multiple vaccinations against SARS-CoV-2 can lead to an increase in the immune response of immunocompromised individuals [8,10,11,12]. So far, there is insufficient data on whether booster vaccination leads to adequate immune responses, especially with regard to the currently predominant SARS-CoV-2 variants delta and omicron.

In the present study, we focused on cellular and humoral immunity to SARS-CoV-2 and its variants in immunosuppressed and immunocompetent vaccinated individuals after at least three mRNA vaccinations. We analyzed cellular immunity by a fluorescence ELISpot assay, which can detect the secretion of IFN-γ and IL-2 simultaneously, as well as by colorimetric SARS-CoV-2-specific IFN-γ and IL-2 ELISpot assays. Neutralizing antibody titers to SARS-CoV-2 D614G (wild type) and its alpha, delta, and omicron variants were analyzed by a cell culture-based neutralization assay. Moreover, potential neutralizing antibodies to variants and mutants of SARS-CoV-2 were determined by competitive fluorescence assay. SARS-CoV-2-specific IgG antibodies were measured by semiquantitative ELISA.

## 2. Materials and Methods

### 2.1. Volunteers

The patient cohort comprised 32 kidney transplant (KTX) patients after booster vaccination against SARS-CoV-2 (Table 1) and without SARS-CoV-2 infection at the timepoint of blood collection. Kidney transplantation was performed at a median of 2 years (range 0.4–11.8) before blood collection. The group included 12 males and 20 females with a median age of 54 years (range 21–76). Of the 32 KTX patients, 31 were vaccinated with Comirnaty^®^ (BioNTech/Pfizer, Mainz, Germany) and one with Spikevax^®^ (Moderna, Cambridge, Massachusetts). Twenty-four of the KTX patients were triple-vaccinated and eight were quadruple-vaccinated. The booster vaccination took place a median of 111 days (range 43–212) before testing. The majority of patients received an immunosuppressive regimen consisting of tacrolimus, mycophenolate, and prednisone. Immunosuppressive therapy was also provided at the time of blood collection and beyond.

We included 17 healthy volunteers after the third vaccination without SARS-CoV-2 infection prior to blood collection as a control group. Of the 17 healthy volunteers, 11 were vaccinated with Spikevax^®^ (Moderna, Cambridge, MA, USA) and six were vaccinated with Comirnaty^®^ (BioNTech/Pfizer, New York, NY, USA). The group consisted of 5 males and 12 females and the median age was 53 years (range 35–65). The cohort was tested at a median of 182 days (range 69–213) after the third vaccination.

This study was approved by the ethics committee of the University Hospital Essen, Germany (20-9753-BO), and all volunteers provided informed consent to participate. It has been performed in accordance with the ethical standards noted in the 1964 Declaration of Helsinki and its later amendments or comparable ethical standards.

### 2.2. CoV-iSpot for Interferon-γ and Interleukin-2

In 31 samples (21 KTX patients, 10 healthy controls), we simultaneously stained for IFN-γ and IL-2 using the CE-marked CoV-iSpot (AID, Strassberg, Germany), as previously described [13]. This fluorescence ELISpot (Fluorospot) contains a peptide mix of the wild type SARS-CoV-2 spike protein. Duplicates of 200,000 peripheral blood mononuclear cells (PBMC) were grown with or without adding the peptide mix (S-pool). The cut-off definition was described previously [14]. We chose 5 as cut-off for positivity for IFN-γ and for IL-2. Among the positive controls, we found an average of 410 spots (range 50–880) in KTX patients for IFN-γ and 463 spots (range 50–1100) for IL-2. In the healthy controls, we found an average of 679 spots (range 486–904) for IFN-γ and 545 spots (range 422–660) for IL-2 in the positive controls.

### 2.3. In-House ELISpot Assay

To further analyze SARS-CoV-2-specific cellular immunity, we used IFN-γ and IL-2 ELISpot assays separately, as previously described [13]. Briefly, 250,000 PBMC of 32 KTX patients and 17 healthy controls were cultured in the presence or absence of either PepTivator^®^ SARS-CoV-2 wild type protein S1/S2, protein S1 (600 pmol/mL of each peptide, Miltenyi Biotec, Bergisch Gladbach, Germany), of the wild type protein S1 (4 µg/mL, Sino Biological, Wayne, PA, USA.) or the omicron variant of the protein S1 (SARS-CoV-2 B.1.1.529, 4 µg/mL, Sino Biological) in 150 µL of AIM-V^®^. Spot numbers were analyzed by an ELISpot reader (AID Fluorospot, Autoimmun Diagnostika GmbH, Strassberg, Germany). The average values of duplicate cell cultures were included. SARS-CoV-2-specific spots were determined as the stimulated minus non-stimulated values (spots increment). We chose a spot increment of 3 for positivity for IFN-γ as well as for IL-2. In the positive controls, we saw on average 432 spots (range 200–600) in KTX patients and 464 spots (range 250–600) in healthy controls for IFN-γ. For IL-2, we saw on average 508 spots (range 200–600) in KTX patients and 517 spots (range 400–600) in healthy controls.

### 2.4. Cells and Viruses

A549-AT cells were cultured in Minimum Essential Medium (MEM) supplemented with 10% fetal calf serum (FCS), 4 mM L-glutamine, 100 IU/mL penicillin, and 100 μg/mL streptomycin at 37 °C and 5% CO_2_. The clinical SARS-CoV-2 isolates D614G (wild type), alpha, delta, and omicron were obtained from nasopharyngeal swabs of COVID-19 patients at our hospital. The SARS-CoV-2 spike gene was sequenced and the corresponding variants were determined after sequence analysis with the WHO list of variants of concern [15]. The viruses were propagated on A549-AT cells and stored at −80 °C. Viral titers were determined using a standard endpoint dilution assay and calculated as 50% tissue culture infective dose (TCID50)/mL, as previously described [16].

### 2.5. Assessment of Neutralizing Antibodies by Cell Culture-Based Neutralization Assay

To assess the neutralizing antibody titers of sera from 28 KTX patients and 11 healthy controls, we used a standard endpoint dilution assay, as described previously [13,17,18]. From the respective sera, serial dilutions (1:20 to 1:2560) were incubated with 100 TCID_50_ of SARS-CoV-2 D614G (wild type), alpha (B.1.1.7), delta (B.1.617.2) or omicron (BA.1) for one hour at 37 °C. Thereafter, the dilutions were added to confluent A549-AT cells [18] in 96-well microtiter plates. After three days of incubation, cells were stained with crystal violet (Roth, Karlsruhe, Germany) solved in 20% methanol (Merck, Darmstadt, Germany). Cells were evaluated for the presence of cytopathic effects (CPE) by light microscopy. The neutralizing titer was defined as the reciprocal of the highest serum dilution at which no CPE was observed in any of the three test wells. A549-AT cells overexpress carboxypeptidase angiotensin-I-converting enzyme 2 (ACE2) receptor and the cellular transmembrane protease serine 2 (TMPRSS2), enabling enhancement of CPE and high SARS-CoV-2 susceptibility. A549-AT cells were cultivated in minimum essential media (MEM), supplemented with 10% (*v/v*) FCS, penicillin (100 IU/mL), and streptomycin (100 µg/mL) at 37 °C in an atmosphere of 5% CO_2_ (all Life Technologies Gibco, Darmstadt, Germany).

### 2.6. Assessment of Neutralizing Antibodies by Competitive Immunofluorescence

For the detection of potential neutralizing antibodies against wild type SARS-CoV-2 and 11 variants of SARS-CoV-2, we used a commercial competitive immunofluorescence assay (Bio-Plex Human SARS-CoV-2 Variant Neutralization Antibody 11-Plex Panel, BIO-RAD, Hercules, CA, USA), as described previously [13]. This competitive immunofluorescence assay works like a binding inhibition assay. Magnetic beats covered with different SARS-CoV-2 spike variants are incubated with soluble, biotin-conjugated ACE2 receptors in the presence of patient sera. Neutralizing serum antibodies compete for binding to the immobilized spike proteins with biotinylated ACE2 receptors. Detection of bound ACE2 receptors is achieved by the addition of streptavidin–phycoerythrin (SA-PE), which binds to the biotinylated ACE2 receptor. The benefit of this method is to detect antibodies that can bind to different mutants and variants of SARS-CoV-2. The upper limit of the system is 1000 ng/mL. We chose 175 ng/mL as the cut-off for positivity, which was defined for a similar testing system [19].

### 2.7. Antibody ELISA

SARS-CoV-2-specific antibodies were detected by a CE-marked Anti-SARS-CoV-2 IgG semiquantitative ELISA (Euroimmun, Lübeck, Germany), according to the manufacturer’s instructions, as described previously [14]. The ELISA plates were coated with wild type recombinant SARS-CoV-2 spike protein (S1 domain). Serum samples were analyzed automatically at a dilution of 1:100, using the Immunomat (Virion\Serion, Würzburg, Germany). An antibody ratio >1.1 was considered positive, ≥0.8 to <1.1 borderline, and <0.8 negative.

### 2.8. Statistical Analysis

Statistical analysis was performed using GraphPad Prism 9.4.0 (San Diego, CA, USA) software. We used Mann–Whitney tests and Spearman correlation to analyze the numerical variables. To compare the categorical variables, we used Fisher’s exact test. Two-sided *p* values < 0.05 were considered significant.

## 3. Results

### 3.1. T Cell Responses in Kidney Transplant Patients and Healthy Volunteers

We examined the cellular immune response in KTX patients and healthy volunteers after booster vaccination and detected significant differences in the commercial CoV-iSpot upon stimulation with the S pool of wild type SARS-CoV-2 (Figure 1). Of the 21 KTX patients, six showed a positive response for IFN-γ, and seven showed a positive response for IL-2. There was a positive reaction only in one KTX patient in the ELISpot measuring simultaneous secretion of IFN-γ and IL-2. Of the 11 healthy controls, seven showed a positive reaction for IFN-γ, seven for IL-2, and two for the simultaneous secretion of IFN-γ and IL-2. The spot increment for IFN-γ and the simultaneous secretion of IFN-γ and IL-2 differed significantly between KTX patients and healthy volunteers (IFN-γ: *p* = 0.005; IFN-γ and IL-2: *p* = 0.001).

Using our in-house ELISpot, we observed in KTX patients versus healthy controls significantly lower numbers of IFN-γ spots after stimulation with S1/S2, S1 or with a recombinantly expressed S1 protein (called S1 Sino hereinafter) (S1/S2: *p* < 0.0001; S1: *p* < 0.0001; S1 Sino: *p* = 0.0005) (Figure 2a,c,f). We also detected significantly lower numbers of IFN-γ spots after stimulation with a recombinant S1 protein of the omicron (B 1.1.529) variant (*p* = 0.0005) (Figure 2g). For IL-2, we could not observe significant differences between KTX patients and healthy volunteers. For IFN-γ, six of the 32 patients displayed a positive reaction towards the S1/S2 peptide mix, seven towards the S1 peptide mix, five to the S1 Sino, and five to the recombinant S1 protein of the omicron variant. For IL-2, 11 of the 32 KTX patients displayed a positive reaction towards S1/S2, 12 towards S1, 13 towards S1 Sino, and 12 to the recombinant S1 protein of the omicron variant. Of the 17 healthy controls, 12 exhibited a positive response to the S1/S2 peptide mix, 15 to the S1 peptide mix, ten to the S1 Sino, and 11 to the recombinant S1 protein of the omicron variant. For IL-2, 11 of the 17 healthy volunteers showed a positive reaction towards S1/S2, 10 towards S1, 5 towards S1 Sino, and 9 to the recombinant S1 protein of the omicron variant. We could not detect significant differences in the cellular immune response between KTX patients after the third vaccination and KTX patients after the fourth vaccination.

Summarizing the cellular data, KTX patients showed significantly lower SARS-CoV-2-specific responses for IFN-γ, but similar mean values for IL-2, compared to healthy controls.

### 3.2. Humoral Immunity in Kidney Transplant Patients and Healthy Controls

We examined the neutralizing antibodies by a cell culture-based neutralization assay and evaluated whether immunocompromised individuals could generate similar levels of neutralizing antibodies against the wild type SARS-CoV-2, alpha variant, delta variant, and omicron (BA.1) variant as the healthy controls. KTX patients showed significantly lower titers of neutralizing antibodies than the healthy controls against all tested variants (wild type: *p* = 0.0001; alpha: *p* = 0.003; delta: *p* < 0.0001; omicron: *p* = 0.0002) (Figure 3). We could not detect significant differences between 24 KTX patients after third vaccination vs. eight KTX patients after fourth vaccination (wild type: *p* = 0.7; alpha: *p* = 0.9; delta: *p* = 0.9; omicron: *p* = 0.6).

We also examined if vaccination can lead to a humoral immune response towards different variants and mutations of SARS-CoV-2 by a competitive immunoassay. KTX patients showed significantly lower concentrations of potential neutralizing antibodies for all tested mutations, namely, alpha, beta, gamma, delta (plus), epsilon, eta, iota, kappa, lambda, mu, and omicron (B 1.1529), compared to the healthy controls (Figure 4). In detail, 18 of the 32 patients responded to the D614G mutation, which can be found in the variants delta and omicron (Figure 4i); 22 of the 32 responded towards the K417N mutation (omicron variant, Figure 4j); and 20 of the 32 showed a positive reaction towards the N501Y mutation (omicron, Figure 4k). All 17 healthy volunteers displayed a positive response towards the D614G mutation, K417N, and N501Y. We detected significant differences between KTX patients and healthy volunteers after booster vaccination in neutralizing antibodies against all variants/mutations examined (*p* < 0.001). The comparison between 24 KTX patients after the third vaccination and eight KTX patients after the fourth vaccination did not display significant differences.

In addition, we measured the antibody ratio in 32 KTX patients and in 17 healthy controls. We detected a significantly lower antibody ratio in KTX patients compared to healthy volunteers (mean ratio of 3.7 vs. 9.3, *p* < 0.0001) (Figure 5). We observed no significant differences between 24 KTX patients after the third vaccination and eight KTX patients after the fourth vaccination (*p* = 0.7).

## 4. Discussion

We observed significant differences in cellular immunity between KTX patients and healthy controls after booster vaccination. While the lower response in the IFN-γ ELISpot was expected, comparable results in the IL-2 ELISpot were at first glance surprising. However, Schrezenmeier et al. found an increase in IL-2-secreting T cells after booster vaccination in KTX patients, whereas the IFN-γ response remained reduced [20]. This is in agreement with our results.

The results of the cell culture based neutralization assay showed comparable mean values of antibodies against the wild type, alpha, and delta, which were moderately decreased in KTX patients as compared to the healthy controls, who had a 1.3-fold higher mean value. For omicron (BA.1), however, differences between KTX patients and healthy controls were more pronounced (*p* < 0.0001). Here, the healthy controls had a 7.8-fold higher mean value of neutralizing antibody titers [21]. This could be due to the fact that these vulnerable groups are still specially protected from possible contact with the virus.

Our results demonstrate that both KTX patients and healthy controls displayed neutralizing antibodies towards variants and mutations of SARS-CoV-2 after booster vaccination against SARS-CoV-2. However, based on the detection of specific antibodies, a protective effect can hardly be assumed. Previous studies have shown that only about 40% of KTX patients develop a humoral immune response after the third vaccination [20,22]. In our study, the measured values were above the cut-off in about 64% of the KTX patients. This could indicate a better response to booster vaccination. An impact of the KTX patients after the fourth vaccination can be excluded, as they do not show any significant differences to KTX patients after the third vaccination.

We detected strongly reduced antibody ratios in KTX patients, which is consistent with the results of previous studies [20,22,23]. However, a study by Bensouna et al. observed an increase in the humoral immune response 30 days after the third vaccination. However, in our study, testing took place at a median of 111 days after vaccination. Other reasons for the lower humoral immune response could be treatment with mycophenolate mofetil or impaired germinal center immunity in immunosuppressed individuals [24].

One limitation of the present study is a lack of data on memory B cells. Notably, other studies showed impaired humoral immunity after mRNA vaccination [25]. Furthermore, it could be demonstrated that a humoral immune response is generated when immunosuppressants are paused [26]. In the cohort studied in our paper, no pausing of immunosuppressive medication was performed. Subsequent studies are needed to comprehensively analyze the memory B cell response in mRNA-vaccinated patients with immunosuppressive treatment.

Our data indicate that there is inadequate immunization in vulnerable groups when compared to healthy controls. In a previous study, we also observed an insufficient humoral immune response in HSCT patients after the third vaccination [13]. Accordingly, other studies of the humoral immune response after SARS-CoV-2 vaccination in vulnerable groups, such as organ transplant and cancer patients, also showed a reduced immune response [27,28,29]. For these individuals, it is recommended to follow all the related safety precautions and to monitor the humoral immune response on a regular basis.

## 5. Conclusions

In conclusion, cellular immunity of KTX patients was significantly lower compared to healthy controls for IFN-γ. For IL-2, KTX patients had a similar mean value of spots increment as the healthy controls. It might be possible that IL-2-secreting T cells also contribute to protection against SARS-CoV-2 infection. However, these cells are not measured by most standard tests. More than half of the KTX patients generated levels of potential neutralizing antibodies to variants of SARS-CoV-2. KTX patients developed neutralizing antibodies, even if they were significantly lower than the titers of healthy controls. Nevertheless, our data suggest that KTX patients are at least partly protected against SARS-CoV-2, either by neutralizing antibodies to variants of SARS-CoV-2 or by cross-reactive T cells.

## Figures and Tables

**Figure 1 vaccines-10-01348-f001:**
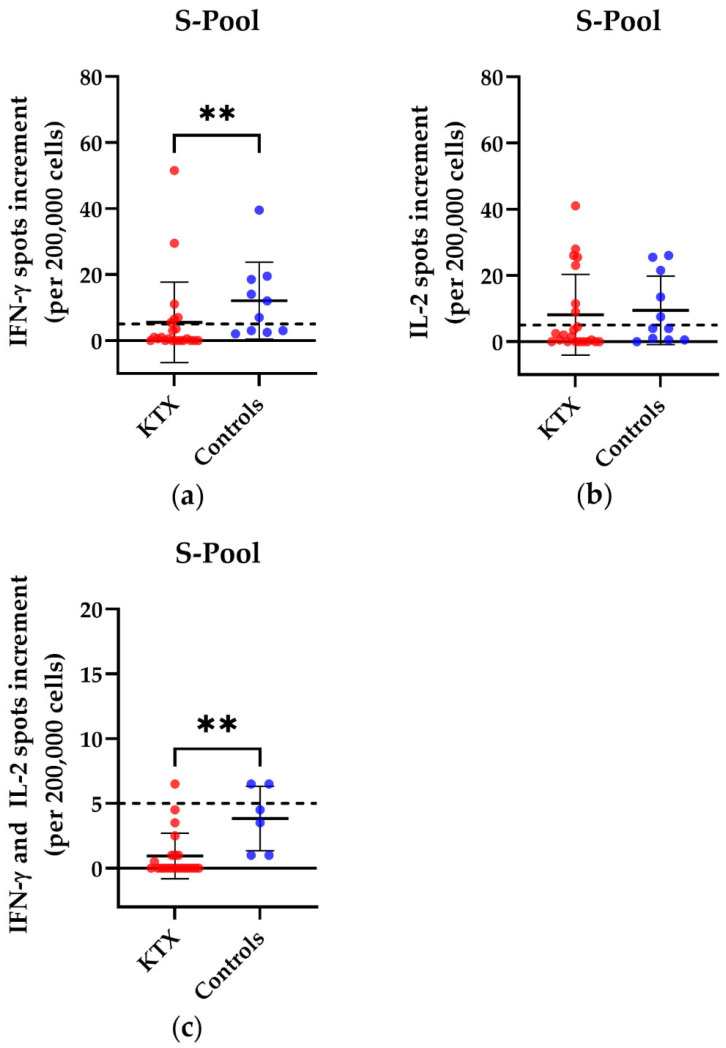
SARS-CoV-2-specific CoV-iSpot responses in kidney transplant (KTX) patients and healthy volunteers after booster vaccination. Distribution of (**a**) IFN-γ, (**b**) IL-2, and (**c**) simultaneous IFN-γ and IL-2 CoV-iSpot responses after stimulation with the S pool of the wild type SARS-CoV-2. Please note the different scales. Red circles show data of the KTX patients, while blue circles indicate data of the healthy volunteers. Two-tailed Mann–Whitney tests were used to compare the responses (** *p* < 0.01). Mean values are represented by horizontal lines, while the standard deviation is represented by error bars. The horizontal line shows the zero line. The dashed line indicates the cut-off.

**Figure 2 vaccines-10-01348-f002:**
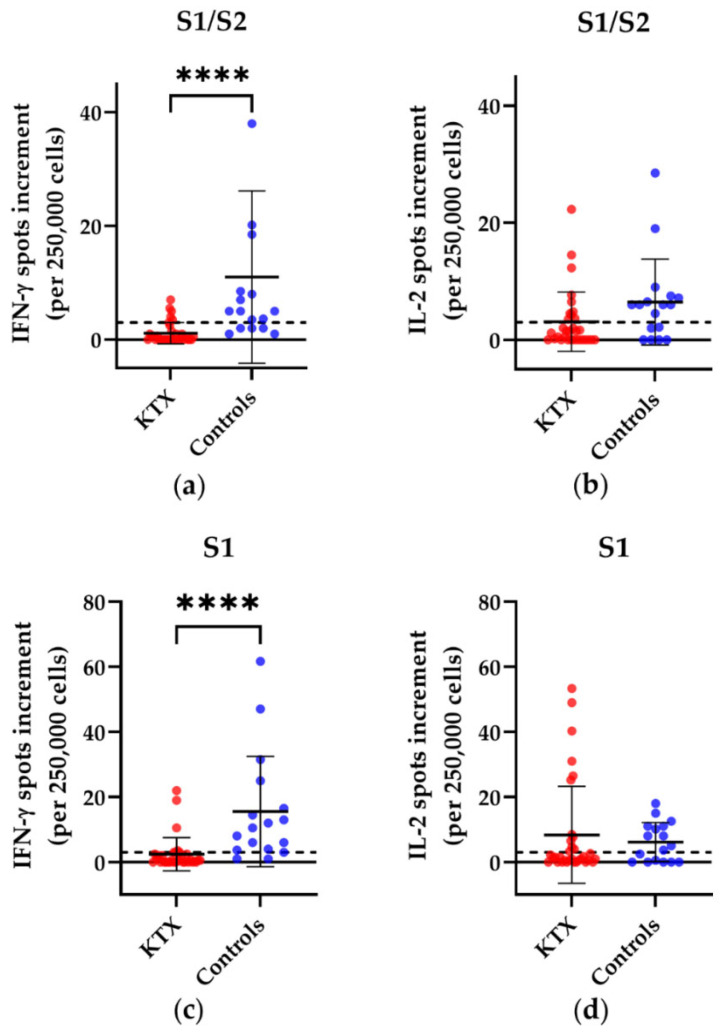
SARS-CoV-2-specific responses in kidney transplant (KTX) patients and healthy controls after booster vaccination, using our in-house ELISpot assay. Distribution of (**a**) IFN-γ and (**b**) IL-2 ELISpot responses after stimulation with an S1/S2 peptide mix, with an S1 peptide mix (**c**,**d**), S1 Sino (**e**,**f**) and S1 Sino of the omicron variant (**g**,**h**). Red circles show data of the KTX patients, while blue circles indicate data of the healthy volunteers. Two-tailed Mann–Whitney tests were used to compare the responses (** *p* < 0.01, *** *p* < 0.001, **** *p* < 0.0001). Mean values are represented by horizontal lines, while the standard deviation is represented by error bars. The horizontal line shows the zero line. The dashed line indicates the cut-off.

**Figure 3 vaccines-10-01348-f003:**
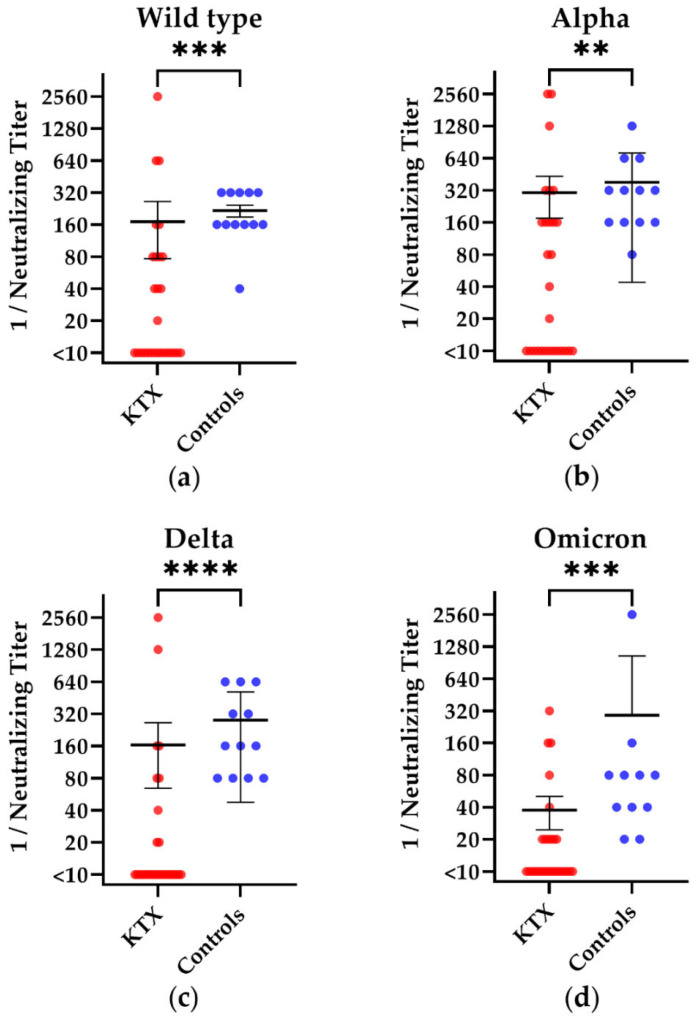
Titer of SARS-CoV-2-specific neutralizing antibodies in kidney transplant (KTX) patients and healthy volunteers. The reciprocal of the titer of neutralizing anti-SARS-CoV-2 (**a**)D614G (wild type), (**b**) alpha, (**c**) delta, and (**d**) omicron (BA.1) antibodies is shown on the *y*-axis. Red circles show data of the KTX patients, while blue circles indicate data of the healthy volunteers. Two-tailed Mann–Whitney tests were used to compare the responses (** *p* < 0.01, *** *p* < 0.001, **** *p* < 0.0001). Mean values are represented by horizontal lines, while the standard deviation is represented by error bars.

**Figure 4 vaccines-10-01348-f004:**
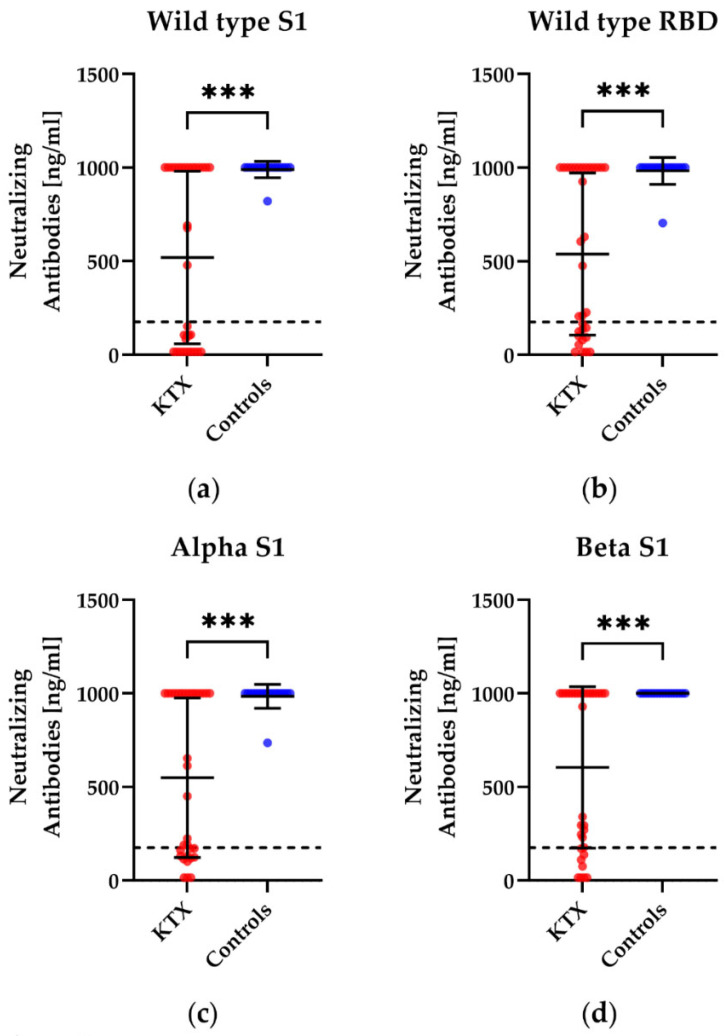
Concentration of potential neutralizing antibodies towards different variants of the subunit 1 of spike protein (S1) or the receptor-binding domain (RBD) of SARS-CoV-2 in kidney transplant (KTX) patients and healthy volunteers (controls) after booster vaccination. Humoral responses after booster vaccination against (**a**) wild type S1, (**b**) wild type RBD, (**c**) alpha S1, (**d**) beta S1, (**e**) gamma RBD, (**f**) E484K RBD, (**g**) epsilon RBD, (**h**) kappa RBD, (**i**) D614G S1, (**j**) K417N RBD and (**k**) N501Y RBD. The mutation D614G can be found in the delta and omicron variants, while K417N and N501Y are mutations in the omicron variant. Red circles show data of the KTX patients, while blue circles indicate the data of healthy volunteers. Two-tailed Mann–Whitney tests were used to compare the responses (*** *p* < 0.001, **** *p* < 0.0001). Mean values are represented by horizontal lines, while the standard deviation is represented by error bars. The dashed line indicates the cut-off.

**Figure 5 vaccines-10-01348-f005:**
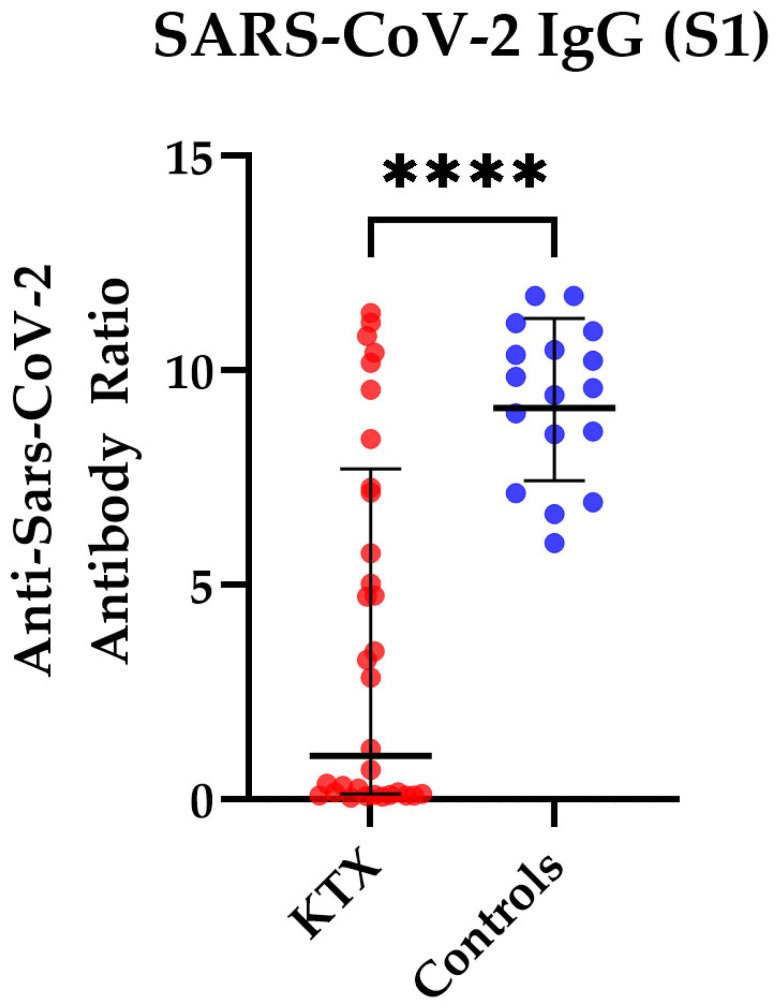
SARS-CoV-2-specific IgG antibody responses in kidney transplant (KTX) patients and healthy controls. SARS-CoV-2-specific IgG antibody responses are shown as antibody ratios, which determines a quotient of antibodies in the patient samples and in a control sample. Red circles show data of the KTX patients, while blue circles indicate the data of healthy volunteers. Two-tailed Mann–Whitney tests were used to compare the responses (**** *p* < 0.0001). Mean values are represented by horizontal lines, while the standard deviation is represented by error bars.

**Table 1 vaccines-10-01348-t001:** Overview of the study cohort.

Characteristics ^1^	Kidney Transplant Recipients	Healthy Controls
sex	12 males 20 females	5 males 12 females
age, y	54 (21–76)	53 (35–65)
tacrolimus	32 (100%)	Ø
mycophenolate	26 (81%)	Ø
belatacept	2 (6%)	Ø
prednisone	32 (100%)	Ø
interval kidney transplantation—blood collection	2 years (0.4–11.8)	Ø
interval vaccination—blood collection	111 days (43–212)	182 days (69–213)

^1^ The data indicate either the median (range) or absolute numbers (percentage). The characteristics of both groups did not differ significantly, as analyzed by Fisher’s exact test (sex: *p* = 0.8) or Mann–Whitney test (age: *p* = 0.5; interval vaccination—blood collection: *p* = 0.1), respectively. Ø: no medication/ no data available.

## Data Availability

The data presented in this study are available on request from the corresponding author. The data are not publicly available due to privacy restrictions.

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
