# Peer review of "Cellular and Humoral Immunity against Different SARS-CoV-2 Variants Is Detectable but Reduced in Vaccinated Kidney Transplant Patients"

_vaccines, 2022, doi:10.3390/vaccines10081348_

Round 1

Reviewer 1 Report

This manuscript describes post-vaccination cellular and humoral immunity in kidney transplant recipients. This topic is very important because this population is at high risk of COVID-19-related death. Despite a small sample size, there is a control group and the methods are valid. The data are interesting and the manuscript is well written.

Comments :

11.  SARS-CoV-2 IgG antibody responses must be expressed in binding antibody units/ml (Methods, results and Figure 5).

22. There was no significant difference after third vaccination and after fourth vaccination in kidney transplant recipients regarding humoral immunity (lines 226, 247 and 264). How many patients were studied in each group ? Quantitative data should be given in the text.

33.  Minor points

-          line 51 : vaccination (rather than inoculation)

-          line 91 : interval kidney transplantation

-          line 280 : reference number

Author Response

Dear Reviewer,

thank you for your comments. These have improved our manuscript.

This manuscript describes post-vaccination cellular and humoral immunity in kidney transplant recipients. This topic is very important because this population is at high risk of COVID-19-related death. Despite a small sample size, there is a control group and the methods are valid. The data are interesting and the manuscript is well written.

Comments :

1. SARS-CoV-2 IgG antibody responses must be expressed in binding antibody units/ml (Methods, results and Figure 5).

Thank you for your comment. The binding antibody titers were assessed in the frame of our routine diagnostics. Therefore, a standard ELISA assay (CE-marked Anti-SARS-CoV-2 IgG semiquantitative ELISA (Euroimmun, Lübeck, Germany)) was used. The results are expressed as a sample/cutoff ratio. Unfortunately, the test gives no values in BAU/ml, and there is no feasible way to calculate BAU/ml values from the ratio values measured. As we used this method to determine binding antibody titers in prior publications (e.g. Lindemann 2021 Transfusion), we would greatly appreciate to keep the results as ratio values. Thank you for your understanding.

2. There was no significant difference after third vaccination and after fourth vaccination in kidney transplant recipients regarding humoral immunity (lines 226, 247 and 264). How many patients were studied in each group? Quantitative data should be given in the text.

We have added the missing information. We examined 24 KTX patients after third and eight KTX patients after fourth vaccination. (p. 10, l. 241-244: We could not detect significant differences between 24 KTX patients after third vaccination vs. eight KTX patients after fourth vaccination (wildtype: p = 0.7; alpha: p = 0.9; delta: p = 0.9; omicron: p = 0.6)., p. 12, l. 263-265: The comparison between 24 KTX patients after third vaccination and eight KTX patients after fourth vaccination did not display significant differences., p. 16, l. 280-282: We observed no significant differences between 24 KTX patients after third vaccination and eight KTX patients after fourth vaccination (p = 0.7).)

3. Minor points

-          line 51 : vaccination (rather than inoculation)

-          line 91 : interval kidney transplantation

-          line 280 : reference number

Thank you, we have updated the manuscript accordingly.

Reviewer 2 Report

Manuscript: vaccines-1853225

Cellular and humoral immunity against different SARS-CoV-2 variants is detectable but reduced in vaccinated kidney transplant patients

Summary: In the submitted manuscript, Thümmler et al. evaluated kidney transplant recipients for evidence of cellular or humoral immunity following a 3rd or 4th COVID-19 booster vaccination and compared the observed responses to healthy controls. Specifically, the authors evaluated cellular immunity using a two-color FluoroSpot assay and measured IFN-γ and IL-2 secretion following stimulation with a mixture of peptides representing the SARS-CoV-2 Spike protein. Moreover, additional single-color ELISPOT assays were also performed using an alternative S1/S2 peptide pool or recombinantly expressed S1 proteins representing wildtype or variant (Omicron) SARS-CoV-2 strains. To characterize humoral immunity, the authors evaluated neutralizing activity of patient serum against a panel of clinical SARS-CoV-2 isolates representing emerging variants. Furthermore, the authors also utilized a multiplexed competitive immunofluorescence assay and a semi-quantitative ELISA to further characterize circulating antibody reactivity against the SARS-CoV-2 Spike protein representing the vaccine strain, along with emerging antigenic variants. Overall, the presented data suggest an impaired vaccine response in kidney transplant recipients as a group relative to healthy controls. Yet, many of the kidney transplant recipients were capable of mounting productive immune responses. Additionally, the data did not suggest an enhancement of immune reactivity in patients that received a 4th COVID-19 booster vaccination.   

Reviewer Comments:

1.       Details regarding the month of collection and SARS-CoV-2 infection status at time of blood donation is not provided.

2.       Despite investigation of T cell and serum antibody reactivity against the SARS-CoV-2 Spike antigen, a weakness in the present manuscript is the lack of memory B cell data. As this component of the humoral immune response is capable of engaging into rapid and robust (anamnestic) response upon antigen re-encounter, such as a breakthrough SARS-CoV-2 infection, assessment of this immune parameter would substantially improve the manuscript. Although characterization of COVID-19 vaccine-elicited B cell reactivity in kidney transplant recipients has previously been reported (PMID 34131023), the time-point investigated in this published work was ideal for assessment of the plasmablast response but did address whether antigen-specific memory B cells were maintained longer-term. As such, the samples collected in the submitted manuscript are ideally suited for addressing this relevant question.

3.       Due to relatively small responses relative to unstimulated control wells detected in the FluoroSpot or ELISPOT assays inclusion of an additional control such as a third-party antigen (seasonal influenza) to verify the functionality of the PBMC input would improve the resolution of the data and allow distinction between a global vs vaccine-specific cellular immune impairment in the kidney transplant recipient cohort.

4.       As kidney and other solid organ transplant recipients have an increased incidence of HGG (hypogammaglobulinemia), especially when treated with mycophenolate mofetil, it is not surprising that the serum antibody response elicited through COVID-19 vaccination would be reduced quantitatively in kidney transplant recipients relative to healthy controls. However, it remains to be determined in the literature (or in this manuscript) whether the antibodies (and memory B cells) elicited in kidney transplant recipients are equally potent qualitatively. More specifically, do B cells undergo successful affinity maturation and selection of clones that target relevant (neutralizing) epitopes. Namely, would kidney transplant recipients still exhibit reduced serum neutralizing activity if the abundance of input IgG was normalized?  

Minor Comment:

The competitive fluorescence binding assay reported in Figure 4 yielded little resolution in antibody activity in the healthy control cohort (Alpha variant as an example), whereas a larger spectrum of antibody “competitiveness” was observed for the kidney transplant cohort. However, in the live virus assay detailed in Figure 3 a larger spectrum of neutralizing titer against the Alpha variant virus was observed in the healthy control cohort and only a subset of kidney transplant recipients possessed neutralizing titers > 1:20. As such, the data generated in these respective assays are not in agreement and should be reconciled. Specifically, what value does the multiplexed competitive fluorescence binding assay provide if it does not correlate with neutralizing activity?

Author Response

Dear Reviewer,

Thank you for your comments. These have improved the manuscript.

Manuscript: vaccines-1853225

Cellular and humoral immunity against different SARS-CoV-2 variants is detectable but reduced in vaccinated kidney transplant patients

Summary: In the submitted manuscript, Thümmler et al. evaluated kidney transplant recipients for evidence of cellular or humoral immunity following a 3rd or 4th COVID-19 booster vaccination and compared the observed responses to healthy controls. Specifically, the authors evaluated cellular immunity using a two-color FluoroSpot assay and measured IFN-γ and IL-2 secretion following stimulation with a mixture of peptides representing the SARS-CoV-2 Spike protein. Moreover, additional single-color ELISPOT assays were also performed using an alternative S1/S2 peptide pool or recombinantly expressed S1 proteins representing wildtype or variant (Omicron) SARS-CoV-2 strains. To characterize humoral immunity, the authors evaluated neutralizing activity of patient serum against a panel of clinical SARS-CoV-2 isolates representing emerging variants. Furthermore, the authors also utilized a multiplexed competitive immunofluorescence assay and a semi-quantitative ELISA to further characterize circulating antibody reactivity against the SARS-CoV-2 Spike protein representing the vaccine strain, along with emerging antigenic variants. Overall, the presented data suggest an impaired vaccine response in kidney transplant recipients as a group relative to healthy controls. Yet, many of the kidney transplant recipients were capable of mounting productive immune responses. Additionally, the data did not suggest an enhancement of immune reactivity in patients that received a 4th COVID-19 booster vaccination.  

Reviewer Comments:

  1. Details regarding the month of collection and SARS-CoV-2 infection status at time of blood donation is not provided.

Thanks for the comment. We have added the requested information on the SARS-CoV-2 infection status. Instead of months, we have indicated the difference between vaccination and blood collection in days. (p. 2, l. 74-76: The patient cohort comprised 32 kidney transplant (KTX) patients after booster vaccination against SARS-CoV-2 (Table 1) and without SARS-CoV-2 infection at the timepoint of blood collection., l. 80-81: The booster vaccination took place a median of 111 days (range 43-212) before testing., l. 85-86: We included 17 healthy volunteers after the third vaccination without SARS-CoV-2 infection prior to blood collection as a control group., l. 89-90: The cohort was tested at a median of 182 days (range 69-213) after the third vaccination.)

  1. Despite investigation of T cell and serum antibody reactivity against the SARS-CoV-2 Spike antigen, a weakness in the present manuscript is the lack of memory B cell data. As this component of the humoral immune response is capable of engaging into rapid and robust (anamnestic) response upon antigen re-encounter, such as a breakthrough SARS-CoV-2 infection, assessment of this immune parameter would substantially improve the manuscript. Although characterization of COVID-19 vaccine-elicited B cell reactivity in kidney transplant recipients has previously been reported (PMID 34131023), the time-point investigated in this published work was ideal for assessment of the plasmablast response but did address whether antigen-specific memory B cells were maintained longer-term. As such, the samples collected in the submitted manuscript are ideally suited for addressing this relevant question.

Thank you for your comment. We agree with the reviewer, that data on memory B cell are interesting and important to understand the B-cell response to SARS-CoV-2 vaccination. However, in our recent study, we focused on investigating the insufficiently studied T cell responses as well as the functional humoral immune response. We took the importance of B-cell data into account by adding some references and mentioning the lack of the respective investigations as a limitation of our study. (p. 17, l. 321-327: One limitation of the present study is a lack of data on memory B cells. Notably, other studies showed impaired humoral immunity after mRNA vaccination [25]. Furthermore, it could be demonstrated that a humoral immune response is generated when immunosuppressants are paused [26]. In the cohort studied in our paper, no pausing of immunosuppressive medication was performed. Subsequent studies are needed to comprehensively analyze the memory B cell response in mRNA vaccinated patients with immunosuppressive treatment.)

  1. Due to relatively small responses relative to unstimulated control wells detected in the FluoroSpot or ELISPOT assays inclusion of an additional control such as a third-party antigen (seasonal influenza) to verify the functionality of the PBMC input would improve the resolution of the data and allow distinction between a global vs vaccine-specific cellular immune impairment in the kidney transplant recipient cohort.

You are right, that is a good point of distinction. We use phytohemagglutinin as a positive control in our ELISpots. If an insufficiently strong reaction of the cells is observed here, the test will not be evaluated. We have written additional information on the values in the positive controls in the material and method section. (p. 3, l. 109-112: Among positive controls, we found an average of 410 spots (range 50-880) in KTX patients for IFN-γ and 463 spots (range 50-1100) for IL-2. In Healthy controls we found an average of 679 spots (range 486-904) for IFN-γ and 545 spots (range 422-660) for IL-2 in positive controls., l. 125-128: In positive controls, we saw on average 432 spots (range 200-600) in KTX patients and 464 spots (range 250-600) in healthy controls for IFN-γ. For IL-2 we saw on average 508 spots (range 200-600) in KTX patients and 517 spots (range 400-600) in healthy controls.)

  1. As kidney and other solid organ transplant recipients have an increased incidence of HGG (hypogammaglobulinemia), especially when treated with mycophenolate mofetil, it is not surprising that the serum antibody response elicited through COVID-19 vaccination would be reduced quantitatively in kidney transplant recipients relative to healthy controls. However, it remains to be determined in the literature (or in this manuscript) whether the antibodies (and memory B cells) elicited in kidney transplant recipients are equally potent qualitatively. More specifically, do B cells undergo successful affinity maturation and selection of clones that target relevant (neutralizing) epitopes. Namely, would kidney transplant recipients still exhibit reduced serum neutralizing activity if the abundance of input IgG was normalized?

That's an important point you raise. Thank you for your comment. In the present study, we focused on the T-cell responses in mRNA vaccinated patients, and assessed binding and neutralizing antibody titers that are assessed in the frame of routine diagnostics. A comprehensive evaluation of B-cell data including affinity maturation was not the focus of the study (and thus, also not approved by the ethic committee). Although we cannot provide data on your request, we discussed this point accordingly.  (p. 17, 318-320: Other reasons for the lower humoral immune response could be treatment with mycophenolate mofetil or impaired germinal center immunity in immunosuppressed individuals [24].)

Minor Comment:

The competitive fluorescence binding assay reported in Figure 4 yielded little resolution in antibody activity in the healthy control cohort (Alpha variant as an example), whereas a larger spectrum of antibody “competitiveness” was observed for the kidney transplant cohort. However, in the live virus assay detailed in Figure 3 a larger spectrum of neutralizing titer against the Alpha variant virus was observed in the healthy control cohort and only a subset of kidney transplant recipients possessed neutralizing titers > 1:20. As such, the data generated in these respective assays are not in agreement and should be reconciled. Specifically, what value does the multiplexed competitive fluorescence binding assay provide if it does not correlate with neutralizing activity?

The fluorescence assay is based on binding inhibition to single peptides and covers only single mutations within the respective SARS-CoV-2 spike antigen. Therefore, it is well suited to specifically detect antibodies against certain mutants and variants. However, no statement about neutralization is possible, since non-neutralizing antibodies may also bind to the respective epitopes. In the cell culture-based neutralization assay, full virus is used. Full viruses contain a broad range of epitopes including neutralizing epitopes within the Spike receptor binding domain. The full-virus based assay was used to determine the true neutralizing capacity of serum antibodies against SARS-CoV-2. Accordingly, the two tests are difficult to compare and does not necessary correlate with each other. (p. 4, l. 168-169: The benefit of this method is to detect antibodies that can bind to different mutants and variants of SARS-CoV-2.)

Reviewer 3 Report

The aim of the study was to compare the SARS-CoV-2 mRNA vaccine immune response between healthy volunteers and kidney transplant patients receiving the different immunosuppressive  therapy. The manuscript is well written, the introduction is clear, the ethical standards were strongly followed, the results are shown in details, the conclusion is confirmed with results and quite important for epidemiologists and virologists. The manuscript deserves publication in the journal. But the exact mRNA vaccine  was not named in the manuscript. And the method of determination of the variants (see page 3) was not detailed, too.   It should be corrected.

Author Response

Dear Reviewer,

Thank you for your comments. These have improved the manuscript.

The aim of the study was to compare the SARS-CoV-2 mRNA vaccine immune response between healthy volunteers and kidney transplant patients receiving the different immunosuppressive therapy. The manuscript is well written, the introduction is clear, the ethical standards were strongly followed, the results are shown in details, the conclusion is confirmed with results and quite important for epidemiologists and virologists. The manuscript deserves publication in the journal. But the exact mRNA vaccine was not named in the manuscript. And the method of determination of the variants (see page 3) was not detailed, too.   It should be corrected.

Thank you for the comment. We have added the missing information on the mRNA vaccine used and the determination of variants. (p. 2, l. 78-79: Of the 32 KTX patients, 31 were vaccinated with Comirnaty® (BioNTech/Pfizer) and one with Spikevax® (Moderna)., l. 86-88: Of the 17 healthy volunteers, 11 were vaccinated with Spikevax® (Moderna) and six were vaccinated with Comirnaty® (BioNTech/Pfizer)., p. 4, l. 135-137: SARS-CoV-2 Spike gene was sequenced and the corresponding variants were determined after sequence analysis with the WHO list of variants of concern [15].)

Reviewer 4 Report

The Authors have compared cellular an humoral responses in SARS-CoV2 of kidney transplant patients with healthy individuals. This study is important and may have similar implication in patients that are on immunosuppressive drugs in other organ transplant patients. here are my comments:

The authors should discuss some other studies that have been done in this context involving transplant patients or cancer patients that are on immunosuppressive drugs.

The details of immunosuppressive drug regimen in these patients is lacking. It is not clear if these patients are on drugs currently.

The main concern I have with the IFN and IL2 ELISPOT is that authors Used PBMCs. The drugs might affect overall CD4 and CD8 numbers and what is see is the number of these T cells in patients is lower and so are the IFN and IL2 spots. Whether function is affected can't be deduced from these graphs.

The neut titers in healthy individuals are very low.

Author Response

Dear Reviewer,

Thank you for your comments. These have improved the manuscript.

The Authors have compared cellular and humoral responses in SARS-CoV2 of kidney transplant patients with healthy individuals. This study is important and may have similar implication in patients that are on immunosuppressive drugs in other organ transplant patients. here are my comments:

The authors should discuss some other studies that have been done in this context involving transplant patients or cancer patients that are on immunosuppressive drugs.

Thank you for your comment. We have included further studies on vulnerable groups as requested. (p. 17, l. 328-334: Our data indicate that there is an inadequate immunization in vulnerable groups when compared to healthy controls. In a previous study, we also observed an insufficient humoral immune response in HSCT patients after the third vaccination [13]. Accordingly, other studies of humoral immune response after SARS-CoV-2 vaccination in vulnerable groups such as organ transplant and cancer patients also showed a reduced immune response [27-29]. For these individuals it is recommended to follow all the related safety precautions and to monitor the humoral immune response on regular basis.)

The details of immunosuppressive drug regimen in these patients is lacking. It is not clear if these patients are on drugs currently.

Thank you for pointing this out, we have added this information to the methods section. (p. 2, l. 83-84: Immunosuppressive therapy was also provided at the time of blood collection and beyond.)

The main concern I have with the IFN and IL2 ELISPOT is that authors Used PBMCs. The drugs might affect overall CD4 and CD8 numbers and what is see is the number of these T cells in patients is lower and so are the IFN and IL2 spots. Whether function is affected can't be deduced from these graphs.

Thank you for this comment. The cell count may be lowered in some patients due to the immunosuppressants. The ELISpot assay with PBMCs is broadly used to determine the cellular immune response to certain antigens. In our study, the assay was conducted to determine the overall T-cell response against SARS-CoV-2 in mRNA vaccinated individuals undergoing immunosuppressive therapy. Although it is not possible to conclude whether their total number of T-cells may differ between the patients, the data give an overall insight of the T-cell response in these patients, and thus, of the effectivity of vaccination.

The neut titers in healthy individuals are very low.

You are right, the titers in the healthy controls are low. However, it must be taken into account that the vaccination in these individuals took place on average six months ago, which explains the relatively low titers. (p. 2, l. 89-90: The cohort was tested at a median of 182 days (range 69-213) after the third vaccination.)

Round 2

Reviewer 2 Report

The revised manuscript adequately addressed my own and other Reviewer comments.

This manuscript is a resubmission of an earlier submission. The following is a list of the peer review reports and author responses from that submission.